# Therapeutic Physical Exercise Programs in the Context of NASH Cirrhosis and Liver Transplantation: A Systematic Review

**DOI:** 10.3390/metabo13030330

**Published:** 2023-02-23

**Authors:** Marwin A. Farrugia, Sebastien Le Garf, Andrea Chierici, Thierry Piche, Philippe Gual, Antonio Iannelli, Rodolphe Anty

**Affiliations:** 1Digestive Center, Centre Hospitalier Universitaire, Archet 2 Hospital, Université Côte d’Azur, 06000 Nice, France; 2CSO PACA-Est, INSERM, C3M, Université Côte d’Azur, CEDEX 3, 06000 Nice, France; 3Centre Hospitalier Universitaire de Nice—Digestive Surgery and Liver Transplantation Unit, Archet 2 Hospital, Université Côte d’Azur, 06000 Nice, France; 4Centre Hospitalier Universitaire, INSERM, U1065, C3M, Université Côte d’Azur, 06000 Nice, France; 5INSERM, U1065, C3M, Université Côte d’Azur, 06000 Nice, France; 6Centre Hospitalier Universitaire de Nice—Digestive Surgery and Liver Transplantation Unit, Archet 2 Hospital, INSERM U1065, Team 8 “Hepatic complications of obesity and alcohol”, Université Côte d’Azur, 06000 Nice, France

**Keywords:** physical activity, NAFLD, NASH cirrhosis, liver transplant recipients, type 2 diabetes

## Abstract

In recent years, various physical exercise interventions have been developed with a view to reducing comorbidity and morbidity rates among patients with chronic diseases. Regular physical exercise has been shown to reduce hypertension and mortality in patients with type 2 diabetes. Diabetes and obesity are often associated with the development of nonalcoholic fatty liver disease, which can lead to liver fibrosis and then (in some cases) nonalcoholic steatohepatitis cirrhosis. We searched the literature for publications on personalized physical exercise programs in cirrhotic patients before and after liver transplantation. Eleven studies in cirrhotic patients and one study in liver transplant recipients were included in the systematic review, the results of which were reported in compliance with the preferred reporting items for systematic reviews and meta-analyses guidelines. The personalized physical exercise programs lasted for 6 to 16 weeks. Our review evidenced improvements in peak oxygen consumption and six-minute walk test performance and a reduction in the hepatic venous pressure gradient. In cirrhotic patients, personalized physical exercise programs improve quality of life, are not associated with adverse effects, and (for transplant recipients) might reduce the 90-day hospital readmission rate. However, none of the literature data evidenced reductions in the mortality rates before and after transplantation. Further prospective studies are needed to evaluate the benefit of long-term physical exercise programs in cirrhotic patients before and after liver transplantation.

## 1. Introduction

### 1.1. Sedentary Lifestyles and Physical Inactivity

Although the term “sedentary” has long been used to describe both physical inactivity in general and time spent sitting in particular, it is important to distinguish clearly between these two concepts. In fact, a person is considered to be active if he/she does more than 150 min of physical activity (PA) per week (wk). However, if that person sits for more than seven hours a day, he or she is considered to be sedentary. Two factors must be considered with regard to a sedentary lifestyle: the total time spent sitting and sitting for a prolonged period. Sitting for more than seven hours a day and sitting for more than two hours in a row have detrimental effects on health [1]. A sedentary lifestyle and physical inactivity have an independent negative effect on health. Human metabolism is designed for PA and muscles use energy from nutrients when the person moves. Promoting regular PA and limiting sedentary behavior are major policy issues for the next decade. Physical inactivity is detrimental to health: according to recent epidemiological data, physical inactivity is responsible for 7.2% of all-cause deaths (i.e., more than four million per year worldwide) [2]. Hence, appropriate public health policies must be implemented.

Physical activity is defined as “any bodily movement produced by the contraction of skeletal muscles leading to an increase in energy expenditure compared to the energy expenditure at rest” [3]. In a recent meta-analysis of 16 studies covering a total of more than one million adults [4], Ekelund et al. divided the cohort into four groups based on the daily time spent sitting (less than four hours, four ot six hours, six to eight hours, or more than eight hours a day). The cohort was also divided into four PA quartiles: <2.5 metabolic equivalent tasks (METs)-h/wk, equivalent to five minutes of moderate-intensity exercise per day; 16 MET-h/week, equivalent to 25–35 min of moderate-intensity exercise per day; 30 MET-h/wk, equivalent to 50–65 min of moderate-intensity exercise per day; and >35.5 MET-h/wk, equivalent to 60–75 min of moderate-intensity exercise per day. These quartiles were compared with two reference groups (the least seated group and the most active group) with regard to the causes of death. The meta-analysis’ main results showed that compared with the reference groups, mortality during follow-up was up to 60% higher in the least active group. In the group that met the World Health Organization’s recommendations for PA, the risk of mortality was still 30% higher for the most sedentary group. It is therefore necessary to combat both a sedentary lifestyle and physical inactivity. In this respect, appropriate PA interventions (delivered by a professional coach or a healthcare professional) might constitute a personalized therapeutic tool.

### 1.2. Physical Activity in People with Type 2 Diabetes

Physical activity is recommended in most chronic diseases and particularly in the management of type 2 diabetes. In a population of non-diabetic patients with impaired glucose tolerance (fasting blood glucose level <1.25 g/L, and then a level between 1.4 and 2 g/L two hours after an oral glucose tolerance test), PA reduced the risk of developing diabetes by 58%, whereas metformin reduced it by only 31% [5]. Glycemia control can be improved by aerobic exercise training; the latter induces a significant decrease (1 to 2%) in hemoglobin A1c (HbA1c) levels due to a reduction in insulin resistance. Furthermore, regular PA has been shown to improve β-cell function [6], insulin sensitivity [7], and vascular function [8]. All these changes reduce the symptoms and complications of diabetes. The latest 2022 consensus statement from the American College of Sports Medicine notes that regular aerobic exercise improves glycemic management in adults with type 2 diabetes [9]. The consensus statement noted a decrease in the daily hyperglycemic time and a 0.5 to 0.7% reduction in overall blood glucose levels, as measured by the level of HbA1c. In people with type 2 diabetes, a moderately high volume of exercise (~500 kcal) performed four to five days per week results in a reduction in visceral fat [10]. Regular training also improves insulin sensitivity, increases physical fitness, and reduces lipid and blood pressure levels—even when weight is not lost [11]. Furthermore, a meta-analysis published in 2016 showed that the average pooled risk reduction for type 2 diabetes was 13% (95% confidence interval (CI): 11%, 16%) for a 10 MET h/wk increment in PA [12].

Given that type 2 diabetes is associated with metabolic syndrome and nonalcoholic fatty liver disease (NAFLD, recently renamed as metabolic dysfunction-associated fatty liver disease), we hypothesized that regular PA can improve the management of patients with this chronic condition.

### 1.3. “Personalized Physical Exercise Programs”: Definitions and Principles

The concept of “personalized physical exercise programs” was introduced in Quebec in 1973. This approach (in which a coach or trainer designs a specific, personalized physical exercise program to meet a patient’s needs) is based on evidence-based medicine, multidisciplinary scientific knowledge, teaching methods, and an evaluation of the patient’s medical, psychological and social characteristics, environment, and disease stage. Furthermore, a successful outcome requires a psychological approach so that the patient commits to the therapeutic physical exercise program. The benefits of an appropriate exercise program cannot be assessed in terms of the patient’s physical ability alone: the better the patient–coach relationship, the greater the likelihood of success. It should be noted that pharmacological interventions are outside the scope of therapeutic physical exercise programs.

Therapeutic physical exercise is a care modality linked to the patient’s needs and resources. It can be preventive or curative depending on the individual’s disease severity, functional abilities, and possible chronic diseases, according to a “frequency, intensity, time, type, total volume and progression” model [9]. The therapeutic challenge for the practitioner is to guide the patient towards a state of biopsychosocial well-being and perform PA on his/her own (thus making the individual more active and less sedentary) by (i) knowing the beneficial health effects of PA on health and the harmful effects of physical inactivity and a sedentary lifestyle, (ii) identifying barriers to and levers for regular PA (i.e., the patient’s motivational status), (iii) identifying ways of increasing levels of PA in daily life (the physical environment, social support, etc.), and (iv) knowing the criteria for the safe implementation of a personalized physical exercise program.

### 1.4. The General Benefits of Personalized Physical Exercise Programs

#### 1.4.1. Benefits for Overall Health

Physical activity is associated with several health benefits. Energy homeostasis is favorably influenced by PA, and PA-related energy expenditure can be increased by the adoption of a more active lifestyle. Indeed, skeletal muscle can represent up to 50% of the total body mass. Modulating this mass through PA can increase the total energy expenditure in the long term. In an appropriate nutritional context, a greater muscle mass is associated with better the regulation of blood sugar levels. As mentioned above, skeletal muscle absorbs and metabolizes glucose in the circulation. During physical effort, skeletal muscle is responsible for nearly 90% of glucose use. It is therefore a central element in preventing the risk of insulin resistance linked to a defect in glucose use, as seen in type 2 diabetes [13].

In summary, PA has a major role in the physiological plasticity of skeletal muscle and (by mitigating metabolic disorders) in systemic metabolic flexibility [14]. Along with its metabolic effects, PA has a mechanical, myogenic role that improves locomotion by increasing muscle mass and effort tolerance. Hence, PA is of major interest in people with myopathies, paralysis, and senescence, for example. The different types of muscle fiber are plastic, which allows them to adapt to the stress of physical exercise [15]. This adaptation is directly linked to the increase in type I myofibrils and to the expression of genes whose products are involved in the oxidation of free fatty acids [16,17].

Physical activity also influences the immune response. At present, there is no consensus on the molecular mechanisms underlying the anti-inflammatory effects of PA. Nevertheless, a few potential mechanisms have been evidenced. According to Gleeson et al. [18], the anti-inflammatory effects of PA might be due to (i) a reduction in the amount of visceral adipose tissue, (ii) stimulation of the secretion of anti-inflammatory cytokines (i.e., myokines and exerkines) by muscle contraction during exercise [19,20], and (iii) a reduction in pro-inflammatory signaling pathways (i.e., toll-like receptor-dependent pathways and the nuclear factor-κB/c-Jun NH_2_-terminal kinase pathway). Furthermore, PA is a powerful antioxidant that reduces oxidative stress by enhancing defense systems (i.e., increasing the secretion of antioxidant species and the activation of enzymes involved in the antioxidant response [21]) and thereby neutralizes reactive oxygen species. This beneficial action of PA reduces the activation of pro-inflammatory signaling pathways.

Regular physical exercise also has an osteogenic action. Indeed, exercise helps to optimize bone growth (e.g., an increase in surface bone mineral density) by regulating mechanical factors (e.g., compression force, vibration, and mechanotransduction), hormonal factors (e.g., insulin-like growth factor-1, parathyroid hormone, and growth hormone), and energy-related factors in (for example) patients with osteoporosis [22]. Lastly, PA has a rheological role [23] by regulating the vasodilator system and increasing fibrinolytic capacity and angiogenesis. Hence, if regular PA is maintained for a sufficiently long time (at least three years), the improvement in arterial hypertension persists both at rest and during exercise. The effect of PA is similar or even superior to that of drug monotherapy.

#### 1.4.2. Benefits for the Liver

Physical activity is a non-pharmacological intervention that improves insulin sensitivity by increasing the phosphorylation of the insulin receptor’s tyrosine residues and reducing the inflammatory state [24]. As mentioned above, PA has a potential antioxidant activity because it enhances the synthesis of antioxidant molecules and neutralizes reactive oxygen species. It allows maintenance of the redox status, which limits the occurrence of oxidative stress [25]. Physical activity therefore reduces the level of activation of pro-inflammatory signaling pathways that contribute significantly to metabolic and/or inflammatory diseases.

It is important to combat lipotoxicity and slow disease progression in patients with NAFLD. Mixed PA (i.e., a combination of aerobic exercise and muscle strengthening) significantly reduces the intrahepatic triglyceride content (i.e., limiting de novo triglyceride synthesis) and free fatty acid levels [26]. Furthermore, personalized physical exercise programs might limit the progression of the low-grade inflammatory state seen in NAFLD. Indeed, PA induces a reduction in the infiltration of pro-inflammatory M1 macrophages, which might be associated with a decrease in levels of pathogen-associated molecules (i.e., danger-associated and pathogen-associated molecular patterns) and in the release of inflammatory cytokines (e.g., interleukin-6, tumor necrosis factor-α, and interferon γ).

Lastly, personalized physical exercise programs increase the hepatic oxidative capacity by increasing levels of peroxisome proliferator-activated receptor α and AMP-activated protein kinase/peroxisome proliferator-activated receptor γ coactivator-1α/carnitine palmitoyltransferase 1a. These proteins are links in important signaling pathways involved in the oxidation of free fatty acids and mitochondrial efficiency (i.e., mitochondrial biogenesis and mitophagy) [27].

### 1.5. Patients with NASH Cirrhosis: Sarcopenia and a Sedentary Lifestyle

#### 1.5.1. Physical Activity in the Setting of Liver Cirrhosis

Liver cirrhosis is a chronic disease characterized by the destruction and abnormal regeneration of the liver parenchyma [28]. It is the eleventh most common cause of death worldwide [29]. Liver cirrhosis is mainly associated with harmful alcohol consumption, viral hepatitis B or C, and metabolic syndrome related to overweight or obesity [30,31].

Personalized physical exercise has an essential role in patients with various chronic diseases and is recommended in many clinical practice guidelines [9], including those on chronic heart failure, chronic lung disease, and solid organ transplantation [32,33,34,35].

NAFLD is the most prevalent cause of liver disease in many parts of the world, and its incidence continues to rise because of the epidemics of obesity and sedentary lifestyles [36]. It has been estimated that between 25% and 30% of world’s global population has NAFLD [37]. NAFLD represents a continuum of liver damage. Approximately 80% of people with NAFLD have steatosis alone, which is associated with a low risk of hepatic or extra-hepatic progression. However, 20% of people with NAFLD have non-alcoholic steatohepatitis (NASH), which is particularly associated with the development of liver fibrosis. Patients may then develop fibrotic NASH and, in some cases, NASH cirrhosis. The severity of fibrosis is associated with overall mortality and mortality from liver-related causes in many cohorts. At the NASH cirrhosis stage, the occurrence of end-stage liver disease (ESLD) may indicate liver transplantation. NAFLD can also be complicated by the development of hepatocellular carcinoma, which occurs in 60% of cirrhotic livers and 40% of non-cirrhotic livers. Some patients with hepatocellular carcinoma may be selected for liver transplantation [37] (Figure 1). Overall, the frequency of liver transplantation for NAFLD is increasing steadily [38].

The majority of patients with NAFLD engage in little PA and this lack has been linked to an elevated risk of metabolic syndrome and NAFLD [39]. Recent guidelines recommend regular PA in individuals with NAFLD [40,41]. NAFLD has also been linked to an elevated risk of myocardial infarction and stroke, since it corresponds to the accumulation of ectopic fat. Individuals with NAFLD are often overweight or obese and often have abnormal blood pressure, glucose, and lipid values. A recent study of the cardiovascular risk in NAFLD found that these associations did not persist after adjusting for conventional risk factors [42]. However, there is considerable evidence to suggest that liver damage (and particularly fibrosis) is associated with the severity or frequency of cardiovascular risk factors [43,44]. A Swedish cohort (n = 10,422) with liver-biopsy-proven NAFLD was monitored for a median of 13.6 years and compared with a matched control group (n = 46,517). The researchers identified the major adverse cardiovascular events (ischemic heart disease, stroke, congestive heart failure, or cardiovascular mortality), which were correlated with the severity of liver fibrosis in this cohort [45].

A sedentary lifestyle and a lack of PA are associated with metabolic syndrome and NAFLD [46]. Weight loss of 10% or more has been associated with an improvement in all the histological characteristics of NAFLD [47]. Performance of moderate-to-high-intensity PA for 90 to 150 min/wk was associated with a significant decrease in intrahepatic fat [48]. Personalized physical exercise (even when moderate, and regardless of whether it is aerobic or resistance-based) improves peripheral and hepatic insulin sensitivity and is of major benefit to patients with NAFLD [36]. It is important to bear in mind that cirrhotic patients are often deficient in vitamin D. Supplementation (along with dietary guidance) might therefore be beneficial [49].

The European Society of Cardiology’s 2021 guidelines recommend that adults perform at least 150–300 min a week of moderate-intensity PA or 75–150 min of vigorous-intensity PA, or an equivalent combination of the two. In sedentary individuals, a gradual increase in the level of activity is recommended. When older adults or individuals with chronic conditions cannot achieve 150 min of moderate-intensity PA a week, they should be as active as their abilities and conditions allow [50,51,52,53]. PA reduces the risk of many adverse health outcomes and risk factors in all ages and both sexes. There is an inverse relationship between moderate-to-vigorous PA and all-cause mortality, cardiovascular morbidity and mortality, and the incidence of type 2 diabetes [53,54].

Physical frailty has also emerged as a critical determinant of mortality in patients with cirrhosis and is the sole predictor of physical robustness after liver transplantation. Given that physical frailty is potentially modifiable by PA, it represents a key target for improving outcomes and quality of life for liver transplant candidates [55] (Figure 1).

#### 1.5.2. Frailty before Liver Transplantation

The concept of frailty is now important in clinical practice; it has a pivotal role in the clinical evaluation prior to liver transplantation, for example. Frailty is defined as “a biological syndrome of decreased reserve and resistance to stressors, resulting from a cumulative decline across multiple physiological systems, and causing vulnerability to adverse outcomes” [56]. This composite variable takes account of functional decline, sarcopenia, malnutrition, physical deconditioning, cognition, balance, cardiopulmonary fitness, walking speed, and muscle strength [56,57,58,59]. Frailty is predictive of mortality, morbidity, hospital admission, transplant delisting, and prolongation of the length of hospital stay and affects 17% to 43% of patients with advanced liver disease [60].

In patients with cirrhosis on the liver transplant waiting list, physical frailty (as measured by the fried frailty index) is a significant predictor of hospital admission and total time spent in hospital per year. Over a 12-month period, frail patients (fried ≥ 3) on the waiting list for liver transplantation were more likely to be admitted to hospital [61]. In 2017, Lai et al. developed a simple frailty index to predict mortality in patients with end-stage liver disease. The frailty index consists of grip strength, chair stands, and balance. When considering the ability to correctly rank patients according to their three-month waiting list mortality risk, the concordance statistic for a combination of the model for end-stage liver disease (MELD) score with sodium with the fried frailty index was 0.82 [62]. Frailty significantly increased the risk of waiting list mortality and/or delisting [58,63]. Recovery after liver transplantation is often complicated, with a 5–10% mortality rate in the early post-transplant period, and this risk is accentuated by frailty [60,64].

It is therefore important to assess frailty in candidates for liver transplantation. A recent literature review identified four tools that can be used for assessment before liver transplantation: the Karnofsky performance status, the liver frailty index, abdominal skeletal muscle mass, and cardiopulmonary exercise testing [65] (Figure 1).

#### 1.5.3. Malnutrition and Sarcopenia

Patients with cirrhosis often present with malnutrition [66], low muscle mass [67,68,69], sarcopenia [70], and impaired health-related quality of life. These factors are associated with a poor prognosis [71].

Sarcopenia is a progressive, generalized skeletal muscle disorder associated with an increased likelihood of an adverse outcome, including falls, fractures, physical disability, and mortality. In its 2018 consensus statement, the European Working Group on Sarcopenia in Older People used low muscle strength as the primary variable in sarcopenia. Specifically, sarcopenia is probable when low muscle strength is detected. The diagnosis of sarcopenia is confirmed by the presence of low muscle quantity or quality [72]. Therefore, sarcopenia is not the same as a lack of PA.

Most patients with a chronic liver disease tend to be physically inactive and have a sedentary lifestyle, which is further complicated by malnourishment. This often results in frailty and sarcopenia, which are significant predictors of elevated morbidity and mortality [35]. The estimated prevalence of sarcopenia in patients with chronic liver cirrhosis is 48.1% (range: 25–70%) [73]. Patients with cirrhosis are known to be less active than the general population and spend 76% of their waking hours in a sedentary state—leading to a high prevalence of frailty [74,75]. Along with an elevated incidence of adverse outcomes prior to transplantation, patients with a low functional capacity also have lower survival rates and longer hospital stays after transplantation [76].

It is important to note that patients with cirrhosis on a waiting list often have severe physical deconditioning and exhibit cirrhosis-induced osteoporosis, malnutrition, fatigue, skeletal muscle abnormalities, and poor overall health that does not improve in the first few weeks after transplantation [77].

Malnutrition in these patients (which usually corresponds to “undernutrition”) results from a lack of food intake and leads to a change in body composition (with a decrease in the fat free mass) and body cell mass, diminished physical and mental functions, and poor clinical disease outcomes [49].

Malnutrition is common in patients with cirrhosis and worsens with liver dysfunction. Sarcopenia is an independent prognostic factor of mortality in this population and should therefore be assessed. Diagnosis of malnutrition may be difficult in patients with cirrhosis and impaired liver function because of fluid variations (ascites and oedema) or the presence of sarcopenic obesity (also referred to as sarcobesity [78]). Indeed, sarcobesity is a combination of obesity (defined as a high percentage of body fat) and sarcopenia (defined as a low skeletal muscle mass and a low level of muscle function). It must be considered as a unique clinical entity that differs from obesity alone or sarcopenia alone. The condition is diagnosed in two steps. Firstly, an impairment in skeletal muscle strength (e.g., handgrip strength or the chair stand test) is evidenced. Secondly, a change in body composition is recorded, i.e., an increase in fat mass and a reduction in muscle mass, assessed as the weight-adjusted appendicular lean mass using dual X-ray absorptiometry or as the weight-adjusted total skeletal muscle mass using bioelectrical impedancemetry [79].

The mid-arm muscle circumference and handgrip strength have shown convincing results in sarcopenia screening. The reported dietary intake, frailty scales, and global assessments can be used to complete the evaluation. Computed tomography assessment of the psoas muscle area in the L3 region appears to be the most relevant technique for confirming sarcopenia [49]. However, these tools have yet to be validated specifically in patients with cirrhosis and diabetes.

In combination with nutritional supplementation, personalized physical exercise programs will be an important weapon in the fight against malnutrition and sarcopenia [49] (Figure 1). Taken as a whole, the literature data suggest that the implementation of personalized physical exercise programs is associated with lower mortality and morbidity rates both before and after liver transplantation.

The objective of the present study was to systematically review the literature on whether cirrhotic patients gain benefit from therapeutic physical exercise programs prior to liver transplantation.

## 2. Materials and Methods

We systematically reviewed the literature and reported the results in compliance with the preferred reporting items for systematic reviews and meta-analyses guidelines [80] (Figure 2). Three investigators (MF, SLG, and AC) screened the Embase, PubMed, Cochrane Library, and Scopus databases up until 31 October 2022 (PROSPERO ID CRD42023392292). After initial screening of the titles and abstracts, the full texts of the selected articles were obtained from the library at the University of Nice (Nice, France) or by contacting the authors. To identify additional publications, we also screened the reference sections of previous reviews and meta-analyses of this topic.

We included comparative and non-comparative studies of the effect of therapeutic physical exercise programs in patients with cirrhosis before liver transplantation or patients who had already undergone liver transplantation. Studies of physical exercise programs in patients with liver disease but not cirrhosis were excluded. Studies assessing how physical performance affects liver disease and/or liver transplantation outcomes without a predefined therapeutic physical exercise program were also excluded. Lastly, case reports and studies conducted on patients under 18 years of age were also excluded. Only full-text records in English were included.

The eligibility of each study was defined according to the population, intervention, comparator, outcome and study design framework, as follows. Population: adult patients (aged ≥18) diagnosed with liver cirrhosis before liver transplantation, or adult patients with a history of liver transplantation; intervention: a therapeutic physical exercise program; comparison: patients not included in a therapeutic physical exercise program (e.g., standard care); outcomes: physical performance, morbidity, and mortality; study design: retrospective or prospective comparative or non-comparative studies.

Mendeley reference software (Mendeley Ltd., London, UK) was used to identify and remove duplicates among the initially identified records. Information on study design/methodology and the participants’ demographic characteristics, baseline characteristics, complications and outcomes were gathered in a computer spreadsheet (Microsoft Excel 2016; Microsoft Corporation, Redmond, WA, USA).

## 3. Results and Discussion

### 3.1. Personalized Physical Exercise Programs in Cirrhotic Patients

Our literature search identified 11 studies in cirrhotic patients and one study in liver transplant recipients (discussed below). Hence, few studies of personalized physical exercise programs in patients with cirrhosis have been performed. None of the selected studies included a program that lasted for more than 16 weeks. Given the absence of longer-term follow-up data, the impact of personalized physical exercise programs on mortality in the setting of liver cirrhosis cannot be assessed reliably. Moreover, the studies’ sample sizes were small.

With regard to liver function, the recent studies (Table 1) showed decreases in hepatic stiffness [35] and the hepatic venous pressure gradient in physically active patients [81,82]. Indeed, Berzigotti et al. showed a significant decrease in HVPG (from 13.9 mmHg to 12.3 mmHg; *p* < 0.0001) after 16 weeks of an intensive lifestyle intervention [82]. The liver frailty index did not change significantly in patients who practiced PA at home [55].

Patients with cirrhosis have poor exercise tolerance when measured objectively (i.e., a low peak oxygen consumption (VO_2_)). A low peak VO_2_ is associated with a shorter survival time, and several studies have demonstrated an independent association between this parameter on one hand and pre- and post-liver transplantation morbidity and mortality rates on the other [77,88,89,90]. Six to eight weeks of supervised aerobic activity significantly increased the peak VO_2_, relative to the pre-exercise baseline value [75,84,85,87]. Relative to dietary advice alone, personalized physical exercise was associated with a smaller proportion of sedentary cirrhotic patients and a significant improvement in the 6MWT in active patients (Table 2) [75,77,83,84].

Anthropomorphic changes were also evidenced in some studies (an increase in the upper thigh circumference and decreases in mid-arm circumference, mid-arm skinfold thickness, thigh circumference [87] and mid-thigh skinfold thickness [86]) but not in others [83,85,91]. Personalized physical exercise programs are associated with a significant decrease in fat mass and an increase in lean mass, as measured by dual energy X-ray [86]. It is noteworthy that a combination of PA and calorie restriction leads to significant weight loss in cirrhotic patients [82]. This might increase survival rates and delay the complications associated with portal hypertension.

Furthermore, the quality of life improved for cirrhotic patients with personalized physical exercise programs [35,55,82,83]. The programs were also associated with a reduction in the level of fatigue [84]. According to a 2018 meta-analysis [91] of four randomized controlled trials that had included cirrhotic patients (regardless of their age, sex, or etiology), a pre-transplantation exercise program (regardless of its supervision status, duration, or intensity) did not cause adverse effects [81,83,84,86]. In particular, there were no significant PA-linked adverse events related to portal hypertension. The meta-analysis did not find an improvement in the Child–Pugh score or MELD score in patients who performed PA, and no significant changes in muscle diameter, 6MWT performance and peak VO_2_ were observed. However, the authors of the meta-analysis noted that the four trials included had small sample sizes and short follow-up periods (only 4 to 12 months).

Physical exercise programs can improve the physical condition of a recipient awaiting liver transplantation, in part, thanks to the establishment of PA with dominant aerobic exercise [85].

Malnutrition is frequent in patients with ESLD; it is observed in >50% of patients with decompensated liver disease and has long been recognized as a determinant prognostic and therapeutic factor [92]. In patients with cirrhosis and a Child–Pugh score <10 and sarcopenia, supplementation with amino acids for six months did not increase muscle mass relative to PA and dietary advice [93]. The combination of supplementation with branched-chain amino acids and a home-based exercise program at the anaerobic threshold for 12 months improved aerobic capacity but did not significantly change the body mass index or biochemical markers of liver function [94]. Moreover, it would be interesting to study the effect of a combination of vitamin D supplementation and personalized physical exercise programs in this population.

Further studies are needed, including prospective assessments of the impact of personalized physical exercise programs on cardiovascular risk factors and anthropometric characteristics.

### 3.2. Personalized Physical Exercise Programs in Liver Transplant Recipients

#### 3.2.1. Liver Transplantation and Cardiovascular Risks

Cardiovascular disease is a major cause of mortality after liver transplantation [95,96]. The cardiovascular event rate among liver transplant recipients is 32% [97]. Ten percent of deaths after transplantation have a cardiovascular cause [98]. These complications can be explained by the patients’ comorbidities prior to transplantation. Indeed, obesity can promote the development of NASH cirrhosis and is also a risk factor for developing cardiovascular complications. The incidence of obesity is increasing [99]: 42.4% of American adults were obese in 2017–2018 [100]. Patients with NAFLD have a greater risk of developing both fatal and non-fatal cardiovascular events (odds ratio = 1.94 in a recent meta-analysis [101]).

Although liver transplantation removes the NASH cirrhosis, it does not correct the underlying etiologic/associated factors: overweight, obesity, and diabetes. On the contrary, the development of metabolic abnormalities and metabolic syndrome is very common after liver transplantation for NASH-related cirrhosis or other types of cirrhosis. In one study, the prevalence of obesity one year after liver transplantation was 24–64% and that of metabolic syndrome was 50–60% [102]. Liver transplantation can exacerbate cardiovascular factors such as diabetes, obesity, hyperlipidemia, and especially hypertension (observed in about 92% of liver transplant recipients [98]).

Diabetes leads to more cardiovascular complications and decreases the five-year survival of transplant patients [103]. Furthermore, some antirejection drugs can enhance the risk of cardiovascular complications. Corticosteroids and calcineurin inhibitors increase the risk of hypertension and diabetes (Table 3) [102].

Physically active patients with NAFLD show lower levels of fibrosis, sarcopenia, and cardiovascular risk factors [104]. Thus, the effective control of cardiovascular risk factors and the prevention of cardiovascular disease through appropriate PA can reduce complications in liver transplant recipients (Figure 1).

#### 3.2.2. Personalized Physical Exercise Programs in Liver Transplant Recipients: A New Promising Approach

Immediately after liver transplantation, patients with malnutrition, sarcopenia, or frailty will need careful management. Dietary supplementation and personalized physical exercise programs will be of value (Figure 1).

An increase in PA improves the wellbeing of organ recipients [105,106]. However, only a few transplant recipients perform PA, notably because of physical limitations, fear of side effects, or associated comorbidities [107,108] such as the cardiovascular risk factors mentioned above.

Liver transplant recipients have a low aerobic capacity, which manifests itself as fatigue after a minor effort [109]. This is explained by prolonged bedrest after transplantation, immunosuppressive drugs (myopathy with steroids), and associated comorbidities (e.g., metabolic syndrome) [36]. Exercise after transplantation can increase lean body mass, muscle strength, and aerobic capacity [110,111]. Moreover, better performance after liver transplantation requires the management of frailty criteria. A recent literature review of eight studies and a total of 1094 patients evaluated the impact of “prehabilitation” on liver transplant candidates. The researchers concluded that prehabilitation can improve the aerobic capacity, with significant improvements in peak VO_2_, 6MWT performance, the liver frailty index, and quality of life. No harmful effects were observed [112].

Our literature review covered a retrospective study by Al-Judaibi et al., the results of which were published in 2019 [113]. Of 458 liver transplant recipients, 258 underwent a comprehensive exercise training program before surgery and 200 did not. The indication for transplantation was a fatty liver in 20.5% of cases. Furthermore, 29% of the liver transplant recipients had diabetes. There were nonsignificant trends toward a lower 90-day hospital readmission rate and a shorter length of stay in the exercise group. Prospective studies are needed to demonstrate the benefit of pre-transplantation PA.

Concerning improvements in functional capacities, two nonrandomized, observational studies of a total of 58 patients evaluated the impact of personalized physical exercise programs: physical capacity improved, as measured by the peak VO_2_ during a 6MWT [114,115]. These findings have been confirmed in controlled trials. In a study of a small number of liver transplant recipients, a 12-week treadmill-based exercise program was associated with a significant improvement in 6MWT performance relative to patients who did not receive this treatment [116]. Similarly, in a randomized trial of 151 liver transplant recipients, individualized exercise and dietary counseling were associated with a significant improvement in the VO_2_ peak relative to patients receiving standard care [110].

Physical activity is an important component of quality of life in liver transplant recipients [117]. The total energy expenditure in transplant recipients is associated with a better quality of life [118]. One to two years after liver transplantation, some patients transplanted for decompensated NASH cirrhosis will show weight (re)gain and metabolic abnormalities, such as type 2 diabetes and metabolic syndrome. As mentioned above, liver transplantation removes the NASH cirrhosis but not overweight, obesity, type 2 diabetes, and metabolic syndrome. This might explain (at least in part) why recurrences of NAFLD and (sometimes) fibrotic NASH can be observed after liver transplantation [119]. It is important to note that metabolic syndrome is a major source of post-liver transplantation morbidity. This condition is common in liver transplant recipients, and the incidence rate is significantly higher one year after transplantation. Metabolic syndrome is associated with lower exercise intensity, independent of age and pre-transplant diabetes status [120]. Therefore, personalized physical exercise programs are essential for increasing aerobic capacity, combating metabolic syndrome (a major cause of morbidity after liver transplantation), and generating the self-confidence that favors a lasting commitment to PA.

Given the high prevalence of obesity and metabolic syndrome after liver transplantation, some patients transplanted for cirrhosis unrelated to NAFLD may develop de novo NAFLD [119]. This can notably be observed in patients who progressively gain weight and (re)develop dysmetabolic complications and/or in patients who were malnourished and/or sarcopenic before liver transplantation.

The above findings highlight the importance of controlling dysmetabolic factors after liver transplantation. Weight and metabolic control must be achieved first and foremost by dietary measures, combined with PA and the avoidance of sedentary behavior (Figure 1). Vitamin D supplementation might be one of the first weight control measures to implement in patients with metabolic syndrome [121]. It would be interesting to evaluate this approach prospectively in liver transplant recipients.

If these dietary approaches are unwanted or ineffective, more aggressive techniques (such as bariatric surgery) can be considered for selected patients [122]. New approaches to weight loss using bariatric endoscopy are promising and need to be evaluated in transplant recipients [123,124]. Lastly, glucagon-like peptide 1 (GLP-1) receptor agonists (e.g., semaglutide), dual GLP-1 and glucose-dependent insulinotropic polypeptide (GIP) receptor agonists (e.g., tirzepatide), dual GLP-1 and glucagon receptor agonists (e.g., cotadutide), or triple glucagon, GIP, GLP-1 receptor agonists (e.g., LY3437943) are new, very effective treatment options [125,126]. Their use in liver transplant recipients will need to be evaluated. In 2016, Rondanelli et al. evaluated the effect of liraglutide (a GLP-1 analogue) in overweight and obese patients with type 2 diabetes [127]. Administration of the drug for 24 weeks at doses of up to 3 mg daily was associated with significantly lower body weight and BMI values. Subjective appetite (as assessed by the Haber score) was also reduced. Furthermore, treatment with liraglutide significantly decreased fat mass, android fat, trunk fat, waist circumference, and the number of risk factors for metabolic syndrome. This might suggest an effect of treatment on hepatic steatosis and a delay in the development of NASH cirrhosis. Further studies are needed to test this hypothesis.

All of these approaches (endoscopy/bariatric surgery and incretin-based drugs) are likely to be synergistic when combined with a personalized physical exercise program and these combinations should be evaluated in liver transplant recipients (Figure 1).

When associated with dietary advice, a personalized physical exercise program initiated before or soon after liver transplantation improves functional capacity and quality of life and reduces risks related to comorbidities. It appears that regular exercise is beneficial for liver transplant recipients, although further long-term studies are needed.

## 4. Conclusions

Personalized physical exercise is a recently developed approach to the management of chronic diseases in general and in patients with cirrhosis and NAFLD. It has beneficial effects on physical abilities and quality of life and is free of undesirable effects. Physical exercise combats frailty in these patients and is essential before and after liver transplantation. Further prospective studies are needed to evaluate the long-term benefit of personalized physical exercise in cirrhotic patients and after a liver transplantation.

## Figures and Tables

**Figure 1 metabolites-13-00330-f001:**
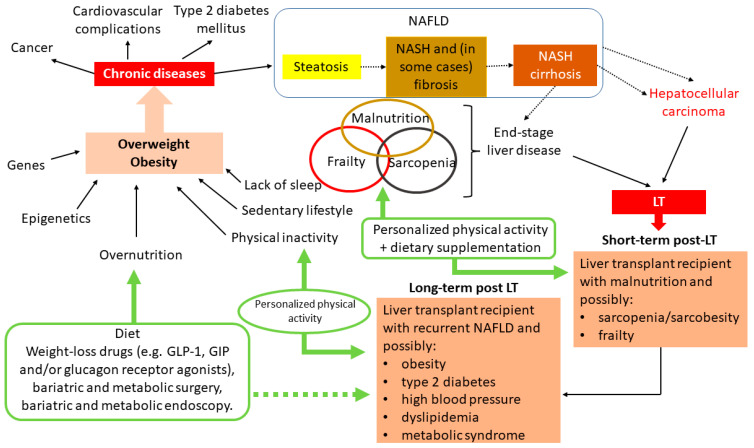
Summary of the possible treatment effects of PA in overweight or obese individuals (particularly those with NASH cirrhosis). **NAFLD**: non-alcoholic fatty liver disease; **NASH**: nonalcoholic steatohepatitis; **GIP**: glucose-dependent insulinotropic polypeptide; **GLP-1**: glucagon-like peptide-1.

**Figure 2 metabolites-13-00330-f002:**
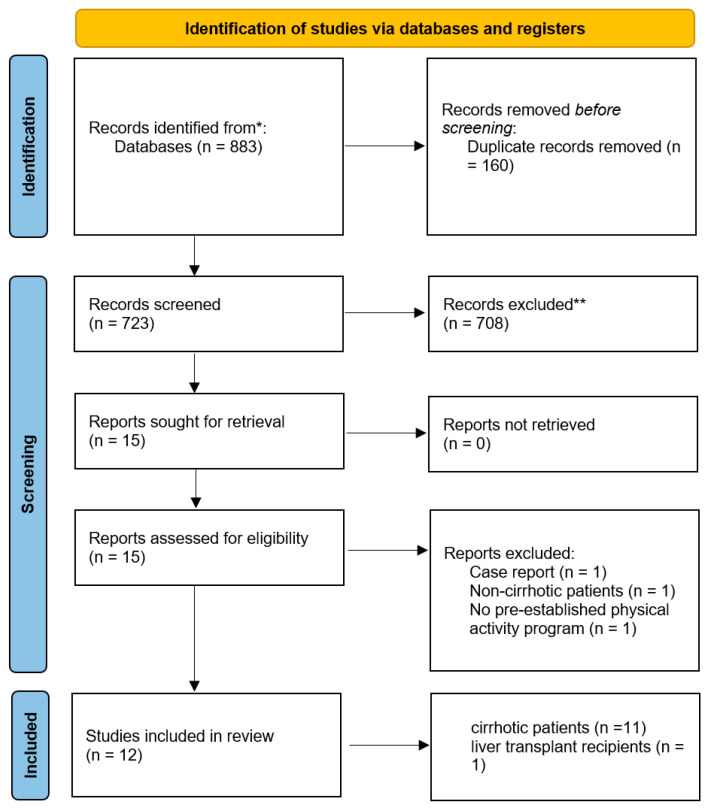
Flow diagram of the systematic review.

**Table 1 metabolites-13-00330-t001:** Baseline characteristics of patients with cirrhosis in the exercise and control groups in studies of personalized physical exercise programs.

Study		Age (Years)	Males (%)	NASH or NAFLD (%)	Child–Pugh Score	MELD	BMI (kg/m^2^)	Diabetes Mellitus	Study Design and Duration	Main Result
Sirisunhirun et al. [35]	HoBET	55.6 (mean)	75%	5%	5 (median)	7.95 (mean)	25.3 (mean)	ND	Randomized controlled trial (12 weeks)	A 12-week home-based exercise training program significantly improved the quality of life for fatigue. However, no significant differences were observed regarding a 6MWT, thigh muscle mass, or hepatic venous pressure gradient.
	Control	57.1 (mean)	55%	0%	5 (median)	7.95 (mean)	25.2 (mean)	ND		
Lai et al. [55]	STRIVE	62 (median)	ND	10%	ND	12 (median)	28 (median)	36%	Multicenter randomized clinical trial (12 weeks)	A structured home strength-training intervention for patients with cirrhosis was associated with a nonsignificant improvement in the LFI and a significant improvement in quality of life.
	Control	61 (median)	ND	16%	ND	13 (median)	28 (median)	24%		
Chen et al. [75]	Exercise	55 (median)	56%	11%	9 (median)	ND	31 (median)	45%	Randomized clinical trial (12 weeks)	A home-based PA program maintained physical performance and improved aerobic fitness according to the 6MWT but not cardiopulmonary exercise testing.
	Control	54 (median)	75%	50%	10 (median)	ND	29 (median)	13%		
Debette-Gratien et al. [77]	Physical activity	51 (mean)	75%	0	7.1 (mean)	12.87 (mean)	ND	ND	Cohort (12 weeks)	A personalized, standardized physical exercise program was acceptable, effective, and safe in patients awaiting liver transplantation. It positively influenced the index of fitness and quality of life.
Román et al. [83]	Exercise	65.5 (mean)	62%	ND	ND	9.5 (mean)	26.7 (mean)	ND	Randomized clinical trial (12 weeks)	A program of moderate physical exercise with leucine supplements for patients with cirrhosis was safe, improved exercise capacity and health-related quality of life, and increased leg muscle mass.
	Control	61.0 (mean)	78%	ND	ND	9.0 (mean)	27.6 (mean)	ND		
Zenith et al. [84]	Exercise	56.4 (mean)	78%	ND	6.2 (mean)	9.7 (mean)	27.7 (mean)	ND	Randomized clinical trial (8 weeks)	Eight weeks of supervised aerobic exercise training increased peak oxygen consumption (VO_2_max) and muscle mass and reduced fatigue in patients with cirrhosis.
	Control	58.6 (mean)	80%	ND	6.3 (mean)	10.2 (mean)	28.9 (mean)	ND		
Morkane et al. [85]	Exercise	55.6 (mean)	87.5%	12.5%	ND	13.7 (mean)	ND	ND	Feasibility study with nonrandomized control cohort (6 and 12 weeks)	Engaging patients with cirrhotic liver disease awaiting liver transplant surgery in an intense, supervised exercise program was safe and feasible.
	Control	55.6 (mean)	82.4%	17.6%	ND	13.2 (mean)	ND	ND		
Román et al. [86]	Exercise	62.0 (mean)	71%	ND	5.4 (mean)	8.2 (mean)	31.5 (mean)	ND	Randomized clinical trial (12 weeks)	A moderate exercise program for patients with cirrhosis led to greater functional capacity, greater muscle mass, and less body fat.
	Relaxation	63.1 (mean)	78%	ND	5.4 (mean)	9.1 (mean)	30.3 (mean)	ND		
Macías-Rodríguez et al. [81]	Exercise	53 (median)	69%	23%	6 (median)	9 (median)	27.5 (mean)	ND	Randomized clinical trial (14 weeks)	A supervised physical exercise program for patients with cirrhosis decreased the HVPG and improved nutritional status, with no changes in the quality of life.
	Control	51 (median)	83%	33%	6 (median)	12 (median)	27.4 (mean)	ND		
Berzigotti et al. [82]	Lifestyle intervention (diet and exercise)	56 (median)	62%	24%	Child A: 92%	9 (median)	33.3 (mean)	42%	Prospective, multicenter, noncontrolled study (16 weeks)	Sixteen weeks of diet and moderate exercise were safe and reduced body weight and portal pressure in overweight/obese patients with cirrhosis and portal hypertension.
Kruger et al. [87]	HoBET	53.0 (mean)	50%	25%	6.35 (mean)	9.05 (mean)	ND	ND	Randomized clinical trial (8 weeks)	Eight weeks of home-based exercise training was an effective therapy for improving peak aerobic power, submaximal aerobic endurance and thigh muscle mass in clinically stable Child—Pugh class A and B cirrhosis.
	Control	56.4 (mean)	65%	25%	6.26 (mean)	9.70 (mean)	ND	ND		

HoBET: home-based exercise training; LFI: liver frailty index; VO_2_max: peak oxygen consumption; 6MWT: six-minute walk test; HVPG: hepatic venous pressure gradient; ND: not determined.

**Table 2 metabolites-13-00330-t002:** Outcomes in studies on interventional physical exercise programs in patients with cirrhosis.

Articles	BMI (kg/m^2^)			Liver Stiffness (kPa)			LFI			6MWT (m)			Peak VO_2_ (mL/kg/min)			MAC (cm)		
	Baseline	End of Study	*p*	Baseline	End of Study	*p*	Baseline	End of Study	Change from Baseline	Baseline	End of Study	*p*	Baseline	End of Study	*p*	Baseline	End of Study	*p*
Sirisunhirun et al. [35]	25.2 (mean)	25.2 (mean)	0.71	14.0 (mean)	12.2 (mean)	**0.016**	ND	ND		479.8 (mean)	498.6 (mean)	0.08	ND	ND		ND	ND	
Lai et al. [55]	ND	ND		ND	ND		3.8 (median)	3.6 (median)	−0.1 (−0.4–0.1)	ND	ND		ND	ND		ND	ND	
Chen et al. [75]	29 (median)	28 (median)	0.98	ND	ND		ND	ND		423 (median)	482 (median)	**0.05**	ND	ND		ND	ND	
Debette-Gratien et al. [77]	ND	ND		ND	ND		ND	ND		480.6 (mean)	520.6 (mean)	**<0.02**	21.5 (mean)	23.2 (mean)	**<0.008**	ND	ND	
Román et al. [83]	26.7 (mean)	27.0 (mean)	> 0.05	ND	ND		ND	ND		365 (median)	445 (median)	**0.01**	ND	ND		27.5 (mean)	28.5 (mean)	>0.05
Zenith et al. [84]	27.7 (mean)	28.0 (mean)	0.17	ND	ND		ND	ND		529.1 (mean)	570.5 (mean)	**0.02**	23.3 (mean)	27.3 (mean)	**0.01**	ND	ND	
Morkane et al. [85]	30.9 (mean)	31.1 (mean)	0.38	ND	ND		ND	ND		ND	ND		16.2 (mean)	18.5 (mean)	**0.02**	35.4 (mean)	35.7 (mean)	0.59
Román et al. [86]	ND	ND		ND	ND		ND	ND		ND	ND		ND	ND		34.1 (mean)	33.5 (mean)	**0.02**
Macías-Rodríguez et al. [81]	27.9 (mean)	28.2 (mean)	0.889	ND	ND		ND	ND		ND	ND		ND	ND		ND	ND	
Berzigotti et al. [82]	ND	ND		ND	ND		ND	ND		ND	ND		ND	ND		ND	ND	
Kruger et al. [87]	29.3 (mean)	29.3 (mean)	0.74	ND	ND		ND	ND		476.0 (mean)	490.7 (mean)	0.08	17.3 (mean)	19.0 (mean)	**0.03**	ND	ND	

LFI: liver frailty index; 6MWT: six-minute walk test; VO_2_: oxygen consumption; MAC: mid-arm circumference; ND: not determined.

**Table 3 metabolites-13-00330-t003:** Cardiovascular side effects of immunosuppressive drugs (adapted with permission from Lucey et al. [102]. Copyright 2023 Wolters Kluwer Health, Inc.).

Adverse Event	Corticosteroids	Calcineurin Inhibitors	mTor Inhibitors	Mycophenolate Mofetil
Kidney injury	-	+++	+ (proteinuria)	-
Hypercholesterolemia	+	+	+++	-
Diabetes	++	+ (tacrolimus)	-	-
Hypertension	+	++	+	-

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
