# Peer review of "Therapeutic Physical Exercise Programs in the Context of NASH Cirrhosis and Liver Transplantation: A Systematic Review"

_metabolites, 2023, doi:10.3390/metabo13030330_

Round 1
Reviewer 1 Report
This paper by Farrugia et al. is a review of the benefits of adapted physical activity on overall health in individuals with various diseases, including those affecting the liver. This is an interesting summary of existing literature and emphasizes the need for non-pharmacological approaches to the treatment of chronic diseases, an area that is understudied. The authors have done an excellent job in structuring this paper and the sections are appropriate and logical. The only recommendation that I have is for the authors to enlist a service to edit for proper English grammar, or to have someone fluent in English provide editing. Some of their points good be made more succinctly and some of the writing is confusing and convoluted. Just an an example, the statements "Thus, a differentiated pedagogy is used to adapt intervention to all patients from a biological, psychological, and social point of view. It is based on the resources, the barriers, and the capacities of the individuals. More precisely, the Adapted PA teacher assesses the functional disabilities but the essential aspect of this initial assessment lies in the pedagogical and educational approach around the psychology of commitment, knowledge in PA for therapeutic purposes. The evaluation of physical abilities cannot alone determine the possibilities of practicing PA. The Adapted PA professional then works on evidence-based medicine. Intervention is therefore not limited to practicing PA but consists in building a set of possibilities, i.e., of practices, relationships, and social participation while theany pharmacological intervention is out of the field of competence of the Adapted PA professional" is quite confusing and could be significantly simplified to allow the reader to understand the exact point the authors are trying to make. This manuscript would be much more "readable" given a thorough overhaul of the writing.
Author Response
Dear reviewer,
We thank you for your comment and for your advice.
We are glad that you enjoyed the review.
We have improved the English grammar to make the article easier to read.
Thanks again for your feedback,
Best regards,
Marwin FARRUGIA

Reviewer 2 Report
This is a really interesting systematic review article on adapted Physical Activity in setting of NASH cirrhosis and 2 liver transplantation.
Despite the paper sounds really interesting and with an high potential of citations I have several suggestions in order to improve the manuscript.
1)the authors must follow the PRISMA guidelines. The term systematic review must me stated into the title
2) the abstract is not informative of this paper. Please report the aim of the review and the main results summarized in table 1
3) the search methodology is missing. Please refer to the prisma guidelines
4) each article summarize in table 1 must report the main results of the study and the study design
5) Line 114 when the authors described that the weight and metabolic control must be achieved first and foremost by dietary measures, please define the important role of the vitamin D3, on the liver and the metabolic pathways, citing this meta analysis by Perna, S., 2019. Is vitamin D supplementation useful for weight loss programs? A systematic review and meta-analysis of randomized controlled trials. Medicina, 55(7), p.368.
6) Discuss the effect of Liraglutide on fatty liver, specifying the action on visceral fat describet by Rondanelly et al and citing this paper. Rondanelli, ., 2016. Twenty-four-week effects of liraglutide on body composition, adherence to appetite, and lipid profile in overweight and obese patients with type 2 diabetes mellitus. Patient preference and adherence, 10, p.407.
Author Response
Dear reviewer,
Thank you for your comment and advice.
We are glad that you found our review interesting and informative.
We have made the changes with PRISMA methodology as requested.
We have completed the article with your references on vitamin D3 and Liraglutide.
Thanks again for your advice,
Best regards,
Marwin FARRUGIA
Translated with www.DeepL.com/Translator (free version)

Reviewer 3 Report
Dear authors,
The topic of the paper by Farruga et al, is very interesting but, is written in such a way that it is difficult to follow and comprehend.
Although the English is quite well, the paper would benefit from some correction by a native speaker.
There are couple of abbreviations that need to be explained, like in lane 38 - MET, lanes 66 and 73 - HbA1c, lane 188 - DAMP and PAMP, and so many more.
Lane 53 - it is said "According to recent epidemiological data, physical inactivity is responsible for 7.2% of deaths from all causes each year, i.e., more than 4 million deaths out of the 56.9 million people who die on average each year [3]" - this sentence doesn't say much, why those deaths occur. Please rephrase that.
Lane 59 - It is said "Adapted PA is the scientific and professional field in which PA is a therapeutic tool." - this needs to be explained more clearly. It is still not clear, even when the whole paragraph, from lane 84 till lane 129 is read, what Adapted PA teacher needs to do. Please rewrite this paragraph.
I would like to ask you to rewrite the entire paper so that it becomes a meaningful scientific paper.
Sincerely,
Iva Lakic
Author Response
Dear reviewer,
Thank you for your comment and advice.
We have improved the English grammar to make the article easier to read and we have detailed the missing abbreviations.
We have completed and modify some paragraphs and resubmit the manuscript soon.
Thanks again for your feedback,
Best regards,
Marwin FARRUGIA

Round 2
Reviewer 1 Report
The authors have satisfactorily addressed my concerns.
Author Response
Thank you for your advice.
Best regards,
Marwin FARRUGIA
Reviewer 3 Report
Dear Authors,
Thank you for the corrected version of your manuscript entitled "Adapted Physical Activity in the setting of NASH cirrhosis and liver transplantation: a systematic review."
Although you have made the suggested changes, the paper would benefit from further corrections.
- The Abstract has been rewritten and is now scientifically better, but the English is still very poor and it is still difficult to understand what the authors were trying to say. Many things are repeated, especially in the Introduction. The paper would really benefit from the help of a native English speaker. If not, the authors should use some of the legally available academic writing assistance programs.
I have a suggestion for the first paragraph of the abstract:
Recently, an adapted physical activity (PA) approach has been developed to reduce comorbidities and morbidity in patients with chronic diseases. Regular PA has been shown to reduce hypertension and mortality in patients with type 2 diabetes. Diabetes and obesity are often associated with the development of Non-Alcoholic Fatty Liver Disease, which can lead to liver fibrosis and eventually Non-Alcoholic steatohepatitis cirrhosis.
- You said that all the abbreviations were explained in the text, but there are still some issues that should be addressed.
- Lane 22 - of the Abstract - it says "Our analysis showed an improvement in the peak VO2, 6-minute walk test..." - the abbreviation is used for VO2, but not for the 6-minute walk test. I would suggest this version: "Our analysis showed an improvement in peak oxygen consumption in the 6-minute walk test...". Be sure to use the abbreviations for the term when it is first mentioned in the paper (for the VO2 - page 13, Table 1, for the 6-minute walk test - page 11, Table 1 (not in the legend of Table 1, and then again on page 19, lane 22);
- lane 43 - the abbreviation for "per week" (wk) needs to be introduced, as it will be used below;
- lane 51 - the abbreviation for physical activity (PA) needs to be introduced in lane 42, where PA is first mentioned;
- lane 94 - it is said "per 10 metabolic equivalent task (MET) h/week increment of PA" - but the abbreviation MET should be earlier in the text.
- The paragraphs in lanes 63-72 should be moved to lane 47 because it talks about PA
- There are still some spelling errors throughout the paper; for example, lane 129 - de-sign, then page 6, lane 223 - it says archeological, but I do not think that was the word the authors intended to use.
- The paragraph about vitamin D3 in the section "7. Adapted Physical Activity in liver transplant recipients" was added, but there is no explanation about the meaning of vitamin D and it’s importance, earlier in the text. This paragraph added in this way has no meaning at all.
Author Response
Dear Authors,
Thank you for the corrected version of your manuscript entitled "Adapted Physical Activity in the setting of NASH cirrhosis and liver transplantation: a systematic review."
Although you have made the suggested changes, the paper would benefit from further corrections.
Dear Reviewer,
Thank you for all your suggestions. You will find our answers below.
- The Abstract has been rewritten and is now scientifically better, but the English is still very poor and it is still difficult to understand what the authors were trying to say. Many things are repeated, especially in the Introduction. The paper would really benefit from the help of a native English speaker. If not, the authors should use some of the legally available academic writing assistance programs.
We followed your advice. We had the English corrected. Thank you also for the suggestions of correction that you brought to us.
We have also modified the term "Adapted physical activity" to make it easier to understand.
We hope this version of the manuscipt will be more readable.
- You said that all the abbreviations were explained in the text, but there are still some issues that should be addressed.
We have filled in missing abbreviations and deleted abbreviations that appeared only once.
- Lane 22 - of the Abstract - it says "Our analysis showed an improvement in the peak VO2, 6-minute walk test..." - the abbreviation is used for VO2, but not for the 6-minute walk test. I would suggest this version: "Our analysis showed an improvement in peak oxygen consumption in the 6-minute walk test...". Be sure to use the abbreviations for the term when it is first mentioned in the paper (for the VO2 - page 13, Table 1, for the 6-minute walk test - page 11, Table 1 (not in the legend of Table 1, and then again on page 19, lane 22);
We have modified according to your request.
- lane 43 - the abbreviation for "per week" (wk) needs to be introduced, as it will be used below;
We have introduced this abbreviation (lane 48).
- lane 51 - the abbreviation for physical activity (PA) needs to be introduced in lane 42, where PA is first mentioned;
We have modified according to your request.
- lane 94 - it is said "per 10 metabolic equivalent task (MET) h/week increment of PA" - but the abbreviation MET should be earlier in the text.
We have modified according to your request.
- The paragraphs in lanes 63-72 should be moved to lane 47 because it talks about PA
We have modified according to your request.
- There are still some spelling errors throughout the paper; for example, lane 129 - de-sign, then page 6, lane 223 - it says archeological, but I do not think that was the word the authors intended to use.
Thanks for the comment, we have changed the mistakes. We hope we haven't missed any.
We meant to say "rheological role" (lane 163).
- The paragraph about vitamin D3 in the section "7. Adapted Physical Activity in liver transplant recipients" was added, but there is no explanation about the meaning of vitamin D and it’s importance, earlier in the text. This paragraph added in this way has no meaning at all
Thank you for your comment, we introduced the concept earlier in the text and simplified this paragraph considerably to make it more consistent with the topic of the review.
We hope you will enjoy this new version of our manuscript.
We thank you again for your guidance in improving our article.
Yours faithfully,
Marwin FARRUGIA
Round 3
Reviewer 3 Report
Dear authors,
Although I tnink that English could be imrpoved, I have no futher comments. I recommend your paper for publishing.
Reviewer